

# Bootstrapping MN and tetragonal CFTs in three dimensions

**Andreas Stergiou**

Theoretical Division, MS B285, Los Alamos National Laboratory, Los Alamos, NM 87545, USA

## Abstract

Conformal field theories (CFTs) with MN and tetragonal global symmetry in $d = 2 + 1$ dimensions are relevant for structural, antiferromagnetic and helimagnetic phase transitions. As a result, they have been studied in great detail with the $\varepsilon = 4 - d$ expansion and other field theory methods. The study of these theories with the nonperturbative numerical conformal bootstrap is initiated in this work. Bounds for operator dimensions are obtained and they are found to possess sharp kinks in the MN case, suggesting the existence of full-fledged CFTs. Based on the existence of a certain large-$N$ expansion in theories with MN symmetry, these are argued to be the CFTs predicted by the $\varepsilon$ expansion. In the tetragonal case no new kinks are found, consistently with the absence of such CFTs in the $\varepsilon$ expansion. Estimates for critical exponents are provided for a few cases describing phase transitions in actual physical systems. In two particular MN cases, corresponding to theories with global symmetry groups $O(2)^2 \rtimes S_2$ and $O(2)^3 \rtimes S_3$, a second kink is found. In the $O(2)^2 \rtimes S_2$ case it is argued to be saturated by a CFT that belongs to a new universality class relevant for the structural phase transition of $NbO_2$ and paramagnetic-helimagnetic transitions of the rare-earth metals Ho and Dy. In the $O(2)^3 \rtimes S_3$ case it is suggested that the CFT that saturates the second kink belongs to a new universality class relevant for the paramagnetic-antiferromagnetic phase transition of the rare-earth metal Nd.

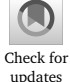

# 1  Introduction and discussion of results

In recent years it has become clear that the numerical conformal bootstrap as conceived in [1][1] is an indispensable tool in our quest to understand and classify conformal field theories (CFTs). Its power has already been showcased in the 3D Ising [3,4,6] and $O(N)$ models [5,7,8], and recently it has suggested the existence of a new cubic universality class in 3D, referred to as $C_3$ or Platonic [9,10]. Now that the method has showed its strength, it is time for it to be applied to the plethora of examples of CFTs in $d = 3$ suggested by the $\varepsilon = 4 - d$ expansion [11–13]. This is of obvious importance, for the bootstrap gives us nonperturbative information that is useful both for comparing with experiments as well as in testing the validity of field theory methods such as the $\varepsilon$ expansion in the $\varepsilon \to 1$ limit.

In this work we apply the numerical conformal bootstrap to CFTs with global symmetry groups that are semidirect products of the form $K^n \rtimes S_n$, where $K$ is either $O(m)$ or the dihedral group $D_4$ of eight elements, i.e. the group of symmetries of the square. These cases have been analyzed in detail with the $\varepsilon$ expansion and other field theory methods due to their importance for structural, antiferromagnetic and helimagnetic phase transitions. This provides ample motivation for their study with the bootstrap, with the hope of resolving some of the controversies in the literature.

One of the cases we analyze in detail in this work is that of $O(2)^2 \rtimes S_2$ symmetry. Such theories are relevant for the structural phase transition of $NbO_2$ and for paramagnetic-helimagnetic phase transitions. Monte Carlo simulations as well as the $\varepsilon$ expansion and the fixed-dimension expansion have been used in the literature. Disagreements both in experimental as well as theoretical results described in [14] and [15] paint a rather disconcerting picture. In this work we observe a clear kink in a certain operator dimension bound—see Fig. 1 below. Following standard intuition, we attribute this kink to the presence of a CFT with $O(2)^2 \rtimes S_2$ symmetry. Using existing results in the literature, namely [5], we can exclude the possibility that this kink is saturated by the CFT of two decoupled $O(2)$ models. Obtaining the spectrum on the kink as explained in [16], we are able to provide estimates for the critical exponents $\beta$ and $\nu$ that are frequently quoted in the literature.[2] We find

$$\beta = 0.293(3), \qquad \nu = 0.566(6). \tag{1}$$

These results suggest that the $\varepsilon$ expansion at order $\varepsilon^4$ perhaps underperforms, for it gives $\beta \approx 0.370$ and $\nu \approx 0.715$ [17, Table II]. Experimental results for $\beta$ for the helimagnet (spiral magnet) Tb are slightly lower and for the helimagnets Ho and Dy higher. Based on the results summarized in [15, Table II] and [12, Table 37] we may estimate that experimentally $\beta = 0.23(4)$, $\nu = 0.53(4)$ for Tb and $\beta = 0.39(4)$, $\nu = 0.57(4)$ for Ho and Dy.

---

[1]See [2] for a recent review.

[2]In terms of the dimensions of the order parameter $\phi$ and the leading scalar singlet $S$ it is $\beta = \Delta_\phi/(3 - \Delta_S)$ and $\nu = 1/(3 - \Delta_S)$.

Also, our result for $\beta$ is below the value measured in the structural phase transition of $NbO_2$, $\beta = 0.40^{+0.04}_{-0.07}$ [18].

Another case of interest is that of CFTs with $O(2)^3 \rtimes S_3$ symmetry. Here we again find a kink—see Fig. 2 below—and for the CFT that saturates it we obtain, with a spectrum analysis,

$$\beta = 0.301(3), \qquad \nu = 0.581(6). \tag{2}$$

Just like in the previous paragraph, we do not find good agreement with results of the $\varepsilon$ expansion, where $\beta \approx 0.363$ and $\nu \approx 0.702$ [17, Table II]. A CFT with $O(2)^3 \rtimes S_3$ symmetry is supposed to describe the antiferromagnetic phase transition of Nd [19–23], but the experimental result for $\beta$ in [24], namely $\beta = 0.36(2)$, is incompatible with our $\beta$ in (2).

In both the $O(2)^2 \rtimes S_2$ and $O(2)^3 \rtimes S_3$ cases we just discussed, we find that the stability of our theory, as measured by the scaling dimension of the next-to-leading scalar singlet, $S'$, is not in question. More specifically, in both cases the scaling dimension of $S'$ is slightly below four, while marginality is of course at three. Therefore, the $\varepsilon$ expansion appears to fail quite dramatically, for it predicts that $S'$, an operator quartic in $\phi$, has dimension slightly above three [17, Table I]. In fact, the purported closeness of the dimension of $S'$ to three according to the $\varepsilon$ expansion has contributed to controversies in the literature regarding the nature of the stable fixed point, with arguments that it may be that of decoupled $O(2)$ models—see section 3 below. The bootstrap shows that the fully-interacting $O(2)^2 \rtimes S_2$ and $O(2)^3 \rtimes S_3$ CFTs are stable.

It is not clear from our discussion so far that our bootstrap bounds are saturated by the CFT predicted by the $\varepsilon$ expansion. This is almost certain, however, as we now explain. Our bootstrap results suggest that there is a well-defined large-$m$ expansion in $O(m)^n \rtimes S_n$ theories. This was verified by the authors of [13] for the fully interacting $O(m)^n \rtimes S_n$ theory of the $\varepsilon$ expansion—see v7 of [13] on the arXiv. The important point here is that since the large-$m$ results of the $\varepsilon$-expansion theory reproduce the behavior we see in our bootstrap bounds, we conclude that the kinks we observe are indeed due to the theory predicted by the $\varepsilon$ expansion.

As we alluded to above, experimental results for phase transitions in the helimagnets Ho and Dy as well as the structural phase transition of $NbO_2$ differ from those in the helimagnet Tb [15]. Prompted by these disagreements, we have explored theories with $O(2)^2 \rtimes S_2$ symmetry in a larger region of parameter space. The idea is that perhaps the helimagnet Tb is not in the same universality class as $NbO_2$ and the helimagnets Ho and Dy, although at criticality both these theories have $O(2)^2 \rtimes S_2$ global symmetry. We find support for this suggestion due to a second kink in our bound and a second local minimum in the central charge—see Figs. 3 and 5 below. Although this kink is not as sharp as the one described above, a spectrum analysis yields

$$\beta = 0.355(5), \qquad \nu = 0.576(8). \tag{3}$$

These numbers are in good agreement with experiments on paramagnetic-helimagnetic transitions in Ho and Dy, $\beta = 0.39(4)$, $\nu = 0.57(4)$ [15, Table II], [12, Table 37] and with the structural phase transition of $NbO_2$, where $\beta = 0.40^{+0.04}_{-0.07}$ [18].

The critical exponent $\beta$ in 2 is not in good agreement with that measured for the antiferromagnetic phase transition of Nd in [24]. Exploration of theories with $O(2)^3 \rtimes S_3$ global symmetry in a larger part of the parameter space reveals a second kink and a second local minimum in the central charge, much like in the $O(2)^2 \rtimes S_2$ case—see Figs. 4 and 6 below. At the second kink we find

$$\beta = 0.394(5), \qquad \nu = 0.590(8), \tag{4}$$

in good agreement with the measurement $\beta = 0.36(2)$ of [24].

The result of our analysis is that there exist two CFTs with $O(2)^2 \rtimes S_2$ symmetry and two CFTs with $O(2)^3 \rtimes S_3$ symmetry. In the $O(2)^2 \rtimes S_2$ case, the first CFT, with critical exponents

given in (1), is relevant for the helimagnet Tb. The second, with critical exponents given in (3), is relevant for the structural phase transition of $NbO_2$ and the helimagnets Ho and Dy. In the case of $O(2)^3 \rtimes S_3$ symmetry we only found experimental determination of the critical exponent $\beta$ in Nd in the literature [24]. It agrees very well with the exponent in (4), computed for the CFT that saturates the second kink. We should mention here that all CFTs appear to have only one relevant scalar singlet, which in experiments would correspond to the temperature. The $\varepsilon$ expansion finds only one CFT in each case, and does not appear to compute the critical exponents and the eigenvalues of the stability matrix with satisfactory accuracy.

Our conclusions do not agree with the suggestion of [15, 25] that in these systems the phase transitions are of weakly first order. The reason for this is that we find kinks in our bootstrap bounds and we suggest that they arise due to the presence of second-order phase transitions. Note that our determinations of the correlation-length critical exponent $\nu$ are in remarkable agreement with experiments. In the cases of Tb and Nd mild tension exists between our results for the order-parameter critical exponent $\beta$ and the corresponding experimental measurements.

For CFTs with symmetry $D_4^n \rtimes S_n$ we have not managed to obtain any bounds with features not previously found in the literature or not corresponding to a symmetry enhancement to $O(2)^n \rtimes S_n$. The $\varepsilon$ expansion does not find a fixed point with $D_4^n \rtimes S_n$ symmetry. The lack of kinks in our plots combined with the lack of CFTs with such symmetry in the $\varepsilon$ expansion suggests that they do not exist in $d = 3$. However, bootstrap studies of $D_4^n \rtimes S_n$ CFTs in larger regions of parameter space are necessary before any final conclusions can be reached.

This paper is organized as follows. In the next section we describe in detail the relevant group theory associated with the global symmetry group $O(m)^n \rtimes S_n$ and derive the associated crossing equation. In section 3 we briefly mention results of the $\varepsilon$ expansion for theories with $O(m)^n \rtimes S_n$ symmetry and some of the physical systems such theories are expected to describe at criticality. In section 4 we turn to the group theory of the global symmetry group $D_4^n \rtimes S_n$ and we derive the crossing equation for this case. In section 5 we mention some aspects of the application of the $\varepsilon$ expansion to theories with $D_4^n \rtimes S_n$ symmetry. Finally, we present our numerical results in section 6 and conclude in section 7.

## 2   MN symmetry

Let us recall some basic facts about semidirect products. To have a well-defined semidirect product, $G = N \rtimes H$, with $N, H$ subgroups of $G$ with $H$ proper and $N$ normal, i.e. $H \subset G$ and $N \lhd G$, we need to specify the action of $H$ on the group of automorphisms of $N$. This action is defined by a map $f : H \to \text{Aut}(N)$, $f : h \mapsto f(h) = f_h$. The action of $f_h$ on $N$ is given by conjugation, $f_h : N \to N$, $f_h : n \mapsto hnh^{-1}$. (By definition $hnh^{-1} \in N$ since $N \lhd G$.) With this definition, $f$ is a homomorphism, i.e. $f_{h_1} f_{h_2} = f_{h_1 h_2}$. One can show that, up to isomorphisms, $N, H$ and $f$ uniquely determine $G$. The multiplication of two elements $(n, h)$ and $(n', h')$ of $G$ is given by

$$(n, h)(n', h') = (n f_h(n'), h h'), \tag{5}$$

the identity element is $(e_N, e_H)$, with $e_N$ the identity element of $N$ and $e_H$ that of $H$, and the inverse of $(n, h)$ is given by

$$(n, h)^{-1} = (f_{h^{-1}}(n^{-1}), h^{-1}). \tag{6}$$

Note that a direct product is a special case of a semidirect product where $f$ is the trivial homomorphism, i.e. the homomorphism that sends every element of $H$ to the identity automorphism of $N$.

In this work we analyze CFTs with global symmetry of the form $K^n \rtimes S_n$, where $K^n$ denotes the direct product of $n$ groups $K$ and $S_n$ the permutation group of $n$ elements. In this

case the action of the homomorphism $f : S_n \to \text{Aut}(K^n)$ is to permute the $K$'s in $K^n$, i.e. $f_\sigma : (k_1, \dots, k_n) \mapsto (k_{\sigma(1)}, \dots, k_{\sigma(n)})$, with $\sigma$ an element of $S_n$ and $k_i$, $i = 1, \dots, n$, an element of the $i$-th $K$ in $K^n$.[3]

The first example we analyze is that of the $MN_{m,n}$ CFT. By this we refer to the CFT with global symmetry $MN_{m,n} = O(m)^n \rtimes S_n$. The vector representation is furnished by the operator $\phi_i$, $i = 1, \dots, mn$. The crucial group-theory problem is of course to decompose $\langle \phi_i(x_1)\phi_j(x_2)\phi_k(x_3)\phi_l(x_4) \rangle$ into invariant subspaces in order to derive the set of crossing equations that constitutes the starting point for our numerical analysis. Invariant tensors help us in this task. As far as the OPE is concerned we have

$$\phi_i \times \phi_j \sim \delta_{ij} S + X_{(ij)} + Y_{(ij)} + Z_{(ij)} + A_{[ij]} + B_{[ij]}, \tag{7}$$

where $S$ is the singlet, $X, Y, Z$ are traceless-symmetric and $A, B$ antisymmetric.

If one thinks of the symmetry breaking $O(mn) \to MN_{m,n}$, then the irreducible representations (irreps) $X, Y, Z$ stem from the traceless-symmetric irrep of $O(mn)$, while $A, B$ stem from the antisymmetric irrep of $O(mn)$. The way to figure out the explicit way the $O(mn)$ representations decompose under the action of the $MN_{m,n}$ group is by constructing the appropriate projectors. The first step to doing that is to construct the invariant tensors of the group under study. This way of thinking, in terms of invariant theory, was recently applied to the $\varepsilon$ expansion in [13], and it turns out to be very useful when thinking about the problem from the bootstrap point of view.

## 2.1 Invariant tensors and projectors

For the $MN_{m,n}$ CFT there are two four-index primitive invariant tensors [13]. They can be defined as follows:

$$\gamma_{ijkl}\phi_i\phi_j\phi_k\phi_l = (\phi_1^2 + \cdots + \phi_m^2)^2 + (\phi_{m+1}^2 + \cdots + \phi_{2m}^2)^2 + \cdots + (\phi_{m(n-1)+1}^2 + \cdots + \phi_{mn}^2)^2, \tag{8a}$$

$$\omega_{ijkl}\phi_i\phi_j\phi_k'\phi_l' = (\phi_1\phi_2' - \phi_2\phi_1')^2 + (\phi_3\phi_4' - \phi_4\phi_3')^2 + \cdots + (\phi_{mn-1}\phi_{mn}' - \phi_{mn}\phi_{mn-1}')^2. \tag{8b}$$

The tensor $\gamma$ is fully symmetric, while the tensor $\omega$ satisfies

$$\omega_{ijkl} = \omega_{jikl}, \qquad \omega_{ijkl} = \omega_{klij}, \qquad \omega_{ijkl} + \omega_{ikjl} + \omega_{iljk} = 0. \tag{9}$$

A non-primitive invariant tensor with four indices is defined by

$$\xi_{ijkl}\phi_i\phi_j\phi_k\phi_l = (\phi_1^2 + \phi_2^2 + \cdots + \phi_{mn}^2)^2, \tag{10}$$

which respects $O(mn)$ symmetry. One can verify that (repeated indices are always assumed to be summed over their allowed values)

$$\gamma_{iijk} = \tfrac{1}{3}(m+2)\delta_{jk}, \qquad \omega_{iijk} = (m-1)\delta_{jk}, \tag{11}$$

and

$$\gamma_{ijmn}\gamma_{klmn} = \tfrac{1}{9}(m+8)\gamma_{ijkl} + \tfrac{2}{27}(m+2)\omega_{ijkl},$$
$$\gamma_{ijmn}\omega_{klmn} = \tfrac{1}{3}(m-1)\gamma_{ijkl} + \tfrac{2}{9}(m+2)\omega_{ijkl},$$
$$\omega_{ijmn}\omega_{klmn} = (m-1)\gamma_{ijkl} + \tfrac{1}{3}(2m-5)\omega_{ijkl},$$
$$\omega_{imjn}\omega_{kmln} = \tfrac{1}{4}(m-1)\gamma_{ijkl} + \tfrac{1}{6}(m+2)\omega_{ijkl} + \tfrac{3}{2}\omega_{ikjl}. \tag{12}$$

---

[3]This type of semidirect product is an example of a wreath product, for which the standard notation is $K \wr S_n$.

With the help of (11) and (12) it can be shown that the tensors

$$P^S_{ijkl} = \tfrac{1}{mn}\delta_{ij}\delta_{kl}, \tag{13a}$$

$$P^X_{ijkl} = \tfrac{1}{m}\gamma_{ijkl} + \tfrac{2}{3m}\omega_{ijkl} - \tfrac{1}{mn}\delta_{ij}\delta_{kl}, \tag{13b}$$

$$P^Y_{ijkl} = (1-\tfrac{1}{m})\gamma_{ijkl} - \tfrac{1}{3}(1+\tfrac{2}{m})\omega_{ijkl}, \tag{13c}$$

$$P^Z_{ijkl} = -\gamma_{ijkl} + \tfrac{1}{3}\omega_{ijkl} + \tfrac{1}{2}(\delta_{ik}\delta_{jl} + \delta_{il}\delta_{jk}), \tag{13d}$$

$$P^A_{ijkl} = \tfrac{1}{3}(\omega_{ijkl} + 2\omega_{ikjl}), \tag{13e}$$

$$P^B_{ijkl} = -\tfrac{1}{3}(\omega_{ijkl} + 2\omega_{ikjl}) + \tfrac{1}{2}(\delta_{ik}\delta_{jl} - \delta_{il}\delta_{jk}), \tag{13f}$$

satisfy

$$P^I_{ijmn}P^J_{mnkl} = P^I_{ijkl}\delta^{IJ}, \qquad \sum_I P^I_{ijkl} = \delta_{ik}\delta_{jl}, \qquad P^I_{ijkl}\delta_{ik}\delta_{jl} = d^I_r, \tag{14}$$

where $d^I_r$ is the dimension of the representation indexed by $I$, with

$$\{d^S_r, d^X_r, d^Y_r, d^Z_r, d^A_r, d^B_r\} = \{1, n-1, \tfrac{1}{2}(m-1)(m+2)n, \tfrac{1}{2}m^2 n(n-1), \tfrac{1}{2}mn(m-1), \tfrac{1}{2}m^2 n(n-1)\}. \tag{15}$$

The dimensions $d^X_r, d^Y_r, d^Z_r$ are as expected from the results of [13, Eq. (5.98)].

Knowledge of the projectors (13a–f) allows the derivation of the corresponding crossing equation in the usual way. The four-point function can be expressed in a conformal block decomposition in the $12 \to 34$ channel as

$$\langle \phi_i(x_1)\phi_j(x_2)\phi_k(x_3)\phi_l(x_4)\rangle = \frac{1}{(x_{12}^2 x_{34}^2)^{\Delta_\phi}} \sum_I \sum_{\mathcal{O}_I} \lambda^2_{\mathcal{O}_I} P^I_{ijkl}\, g_{\Delta_I, \ell_I}(u, v), \tag{16}$$

where the sum over $I$ runs over the representations $S, X, Y, Z, A, B$, $x_{ij} = x_i - x_j$, $\lambda^2_{\mathcal{O}_I}$ are squared OPE coefficients and $g_{\Delta_I, \ell_I}(u, v)$ are conformal blocks[4] that are functions of the conformally-invariant cross ratios

$$u = \frac{x_{12}^2 x_{34}^2}{x_{13}^2 x_{24}^2}, \qquad v = \frac{x_{14}^2 x_{23}^2}{x_{13}^2 x_{24}^2}. \tag{17}$$

The crossing equation can now be derived. With

$$F^\pm_{\Delta,\ell}(u, v) = v^{\Delta_\phi} g_{\Delta,\ell}(u, v) \pm u^{\Delta_\phi} g_{\Delta,\ell}(v, u), \tag{18}$$

---

[4]We define conformal blocks using the conventions of [26].

we find[5]

$$
\sum_{S^+} \lambda_{\mathcal{O}}^2 \begin{pmatrix} F_{\Delta,\ell}^- \\ 0 \\ 0 \\ 0 \\ F_{\Delta,\ell}^+ \\ 0 \end{pmatrix}
+ \sum_{X^+} \lambda_{\mathcal{O}}^2 \begin{pmatrix} -F_{\Delta,\ell}^- \\ F_{\Delta,\ell}^- \\ 0 \\ 0 \\ -F_{\Delta,\ell}^+ \\ F_{\Delta,\ell}^+ \end{pmatrix}
+ \sum_{Y^+} \lambda_{\mathcal{O}}^2 \begin{pmatrix} 0 \\ \frac{m-1}{n} F_{\Delta,\ell}^- \\ F_{\Delta,\ell}^- \\ 0 \\ 0 \\ -\frac{m+2}{2n} F_{\Delta,\ell}^+ \end{pmatrix}
+ \sum_{Z^+} \lambda_{\mathcal{O}}^2 \begin{pmatrix} 0 \\ 0 \\ 0 \\ F_{\Delta,\ell}^- \\ -\frac{1}{2} F_{\Delta,\ell}^+ \\ \frac{1}{2n} F_{\Delta,\ell}^+ \end{pmatrix}
$$

$$
+ \sum_{A^-} \lambda_{\mathcal{O}}^2 \begin{pmatrix} 0 \\ 0 \\ \frac{1}{m} F_{\Delta,\ell}^- \\ 0 \\ 0 \\ \frac{1}{2n} F_{\Delta,\ell}^+ \end{pmatrix}
+ \sum_{B^-} \lambda_{\mathcal{O}}^2 \begin{pmatrix} -F_{\Delta,\ell}^- \\ \frac{1}{n} F_{\Delta,\ell}^- \\ 0 \\ F_{\Delta,\ell}^- \\ \frac{1}{2} F_{\Delta,\ell}^+ \\ -\frac{1}{2n} F_{\Delta,\ell}^+ \end{pmatrix}
= \begin{pmatrix} 0 \\ 0 \\ 0 \\ 0 \\ 0 \\ 0 \end{pmatrix}.
$$
$$(19)$$

The signs that appear as superscripts in the various irrep symbols indicate the spins of the operators we sum over in the corresponding term: even when positive and odd when negative.

## 3 MN anisotropy

The $MN_{m,n}$ fixed points were first studied in [17, 27–30] and more recently in [13, 31]. The relevant Lagrangian is[6]

$$
\mathscr{L} = \tfrac{1}{2} \partial_\mu \phi_i \partial^\mu \phi_i + \tfrac{1}{8}(\lambda \xi_{ijkl} + \tfrac{1}{3} g \gamma_{ijkl}) \phi_i \phi_j \phi_k \phi_l. \tag{20}
$$

In the $\varepsilon$ expansion below $d = 4$ (20) has four inequivalent fixed points. They are

1. Gaussian ($\lambda = g = 0$),

2. $O(mn)$ ($\lambda > 0, g = 0$)

3. $n$ decoupled $O(m)$ models ($\lambda = 0, g > 0$),

4. $n$ coupled $O(m)$ models with symmetry $MN_{m,n} = O(m)^n \rtimes S_n$ ($\lambda \neq 0, g > 0$).[7]

These fixed points are known to be physically relevant for $m = 2$ and $n = 2, 3$. As already mentioned in the introduction, the $MN_{2,2}$ fixed point has been argued to describe the structural phase transition of $NbO_2$ (niobium dioxide) and paramagnetic-helimagnetic transitions in the rare-earth metals Ho (holmium), Dy (dysprosium) and Tb (terbium). The $MN_{2,3}$ fixed point is relevant for the antiferromagnetic phase transitions in $K_2IrCl_6$ (potassium hexachloroiridate), $TbD_2$ (terbium dideuteride) and Nd (neodymium) [19–23].

In the $\varepsilon$ expansion the $MN_{2,2}$ CFT is equivalent to a theory with $O(2)^2/\mathbb{Z}_2$ symmetry [12, 13, 31]. Lagrangians with $O(2)^2/\mathbb{Z}_2$ symmetry have fixed points with collinear (also referred to as sinusoidal) or noncollinear (also referred to as chiral) order [12, 14]. The equivalence of the $MN_{2,2}$ and $O(2)^2/\mathbb{Z}_2$ CFTs in the $\varepsilon$ expansion happens in the collinear region, and so

---

[5]In (19) we omit, for brevity, to label the $F_{\Delta,\ell}$'s and $\lambda_{\mathcal{O}}^2$'s with the appropriate index $I$. The appropriate labeling, however, is obvious from the overall sum in each term.

[6]Compared to couplings $\lambda, g$ of [13, Sec. 5.2.2] we have $\lambda^{\text{here}} = \lambda^{\text{there}} - \frac{m+2}{3(mn+2)} g^{\text{there}}$ and $g^{\text{here}} = g^{\text{there}}$.

[7]Although the theory of $n$ decoupled $O(m)$ models in item 3 on the list also has symmetry $MN_{m,n}$, we will never characterize it that way; we will reserve that characterization for the fully-interacting case in 4.

our results in this work do not apply to stacked triangular antiferromagnets, whose phase transitions are described by $O(2)^2/\mathbb{Z}_2$ CFTs in the noncollinear region. Our results, however, could be of relevance to magnets with sinusoidal spin structures [14].

The stability of the $MN_{m,n}$ fixed point for $m = 2$ and $n = 2, 3$ has been supported by higher-loop $\varepsilon$ expansion calculations [17, 29, 30]. However, there exist higher-loop calculations based on the fixed-dimension expansion—see [12] and references therein—indicating that the stable fixed point is actually that of $n$ decoupled $O(2)$ models. As mentioned in the introduction, our numerical results indicate that the $MN_{2,2}$ and $MN_{2,3}$ theories are both stable.

## 4  Tetragonal symmetry

The tetragonal CFT [12, 13] has global symmetry $R_n = D_4^n \rtimes S_n$, where $D_4$ is the eight-element dihedral group. For $n = 0$ $R_0 = \{e\}$, where $e$ is the identity element, and for $n = 1$ $R_1 = D_4$. The order of $R_n$ is $\mathrm{ord}(R_n) = 8^n n!$. Note that $R_n$ is a subgroup of the hypercubic group $C_N = \mathbb{Z}_2^N \rtimes S_N$, $N = 2n$, whose order is $\mathrm{ord}(C_N) = 2^N N!$. It is easy to see that $\mathrm{ord}(C_N)/\mathrm{ord}(R_n) = (2n - 1)!!$, which is an integer for any integer $n \geqslant 0$.

The number of irreps of the group $R_n$ for $n = 0, 1, 2, 3, 4, 5, \ldots$ is $1, 5, 20, 65, 190, 506, \ldots$, respectively.[8] Among the irreps of $R_n$ one always finds a $2n$-dimensional one; we will refer to this as the vector representation $\phi_i, i = 1, \ldots, 2n$.

In this work we analyze bootstrap constraints on the four-point function of the vector operator $\phi_i$. A standard construction of the character table shows that the group $R_2$ has eight one-dimensional, six two-dimensional and six four-dimensional irreps.[9] In this case we may write[10]

$$\overset{4}{\phi}_i \times \overset{4}{\phi}_j \sim \delta_{ij}\overset{1}{S} + \overset{2}{W}_{(ij)} + \overset{1}{X}_{(ij)} + \overset{2}{Y}_{(ij)} + \overset{4}{Z}_{(ij)} + \overset{2}{A}_{[ij]} + \overset{4}{B}_{[ij]}, \tag{21}$$

where $S$ is the singlet. The dimensions of the various irreps are given by the number over their symbol. $W, X, Y, Z$ are two-index symmetric and traceless, while $A, B$ are two-index antisymmetric.

### 4.1  Invariant tensors

In the tetragonal case there are three primitive invariant tensors with four indices, defined by

$$\delta_{ijkl}\phi_i\phi_j\phi_k\phi_l = \phi_1^4 + \phi_2^4 + \cdots + \phi_{2n}^4, \tag{22a}$$

$$\zeta_{ijkl}\phi_i\phi_j\phi_k\phi_l = 2(\phi_1^2\phi_2^2 + \phi_3^2\phi_4^2 + \cdots + \phi_{2n-1}^2\phi_{2n}^2), \tag{22b}$$

$$\omega_{ijkl}\phi_i\phi_j\phi_k'\phi_l' = (\phi_1\phi_2' - \phi_2\phi_1')^2 + (\phi_3\phi_4' - \phi_4\phi_3')^2 + \cdots + (\phi_{2n-1}\phi_{2n}' - \phi_{2n}\phi_{2n-1}')^2. \tag{22c}$$

The tensors $\delta, \zeta$ are fully symmetric, while the tensor $\omega$ is the same as that in (8b) for $m = 2$. It can be verified that these satisfy

$$\delta_{iijk} = 3\zeta_{iijk} = \omega_{iijk} = \delta_{jk}, \tag{23}$$

---

[8]These numbers have been obtained with the use of the freely available software GAP [32].

[9]Character tables for a wide range of finite groups can be easily generated using GAP [32].

[10]Of course these $S, X, Y, Z, A, B$ have nothing to do with the ones of section (2).

and

$$\delta_{ijmn}\delta_{klmn} = \delta_{ijkl},$$
$$\delta_{ijmn}\zeta_{klmn} = \tfrac{1}{3}\zeta_{ijkl} + \tfrac{2}{9}\omega_{ijkl},$$
$$\delta_{ijmn}\omega_{klmn} = \zeta_{ijkl} + \tfrac{2}{3}\omega_{ijkl},$$
$$\zeta_{ijmn}\zeta_{klmn} = \tfrac{1}{9}\delta_{ijkl} + \tfrac{4}{9}\zeta_{ijkl} - \tfrac{4}{27}\omega_{ijkl}, \tag{24}$$
$$\zeta_{ijmn}\omega_{klmn} = \tfrac{1}{3}\delta_{ijkl} - \tfrac{2}{3}\zeta_{ijkl} + \tfrac{2}{9}\omega_{ijkl},$$
$$\omega_{ijmn}\omega_{klmn} = \delta_{ijkl} + \zeta_{ijkl} - \tfrac{1}{3}\omega_{ijkl},$$
$$\omega_{imjn}\omega_{kmln} = \tfrac{1}{4}\delta_{ijkl} + \tfrac{1}{4}\zeta_{ijkl} + \tfrac{2}{3}\omega_{ijkl} + \tfrac{3}{2}\omega_{ikjl}.$$

These relations are valid for any $n \geqslant 2$.

To verify that there are only three invariant polynomials of $R_n$ made out of the components of the vector $\phi_i$, we have computed the Molien series for $n = 2, 3, 4$.[11] To do this, we think of $R_n$ as represented by $2n \times 2n$ matrices acting on the $2n$-component vector $\phi_i^T$. Using those matrices, which represent the group elements $g_i \in G$ as $\rho(g_i)$, $i = 1, \ldots, \mathrm{ord}(G)$, we can then explicitly compute the Molien series. The Molien formula is

$$M(t) = \frac{1}{\mathrm{ord}(G)} \sum_{i=1}^{\mathrm{ord}(G)} \frac{1}{\det(\mathbb{1} - t\,\rho(g_i))}, \tag{25}$$

where $\mathbb{1}$ is the identity matrix of appropriate size. It is obvious that the summands in (25) only depend on the conjugacy class, so the sum can be taken to be over conjugacy classes with the appropriate weights. For $n = 2, 3, 4$ (25) gives, respectively,

$$M_2(t) = \frac{t^4 - t^2 + 1}{(t^4 + 1)^2(t^2 + 1)^2(t^2 - 1)^4},$$
$$M_3(t) = \frac{t^{16} - t^{14} + t^{12} + t^8 + t^4 - t^2 + 1}{(t^4 + t^2 + 1)^2(t^4 - t^2 + 1)^2(t^4 + 1)(t^2 + 1)^3(t^2 - 1)^6}, \tag{26}$$
$$M_4(t) = \frac{(t^{20} - t^{18} + t^{14} + t^{12} - t^{10} + t^8 + t^6 - t^2 + 1)(t^8 - t^6 + t^4 - t^2 + 1)}{(t^8 + 1)(t^4 - t^2 + 1)(t^4 + 1)^2(t^2 + t + 1)^2(t^2 - t + 1)^2(t^2 + 1)^4(t^2 - 1)^8},$$

whose series expansions are

$$M_2(t) = 1 + t^2 + 3t^4 + 4t^6 + 8t^8 + \mathrm{O}(t^{10}),$$
$$M_3(t) = 1 + t^2 + 3t^4 + 5t^6 + 10t^8 + \mathrm{O}(t^{10}), \tag{27}$$
$$M_4(t) = 1 + t^2 + 3t^4 + 5t^6 + 11t^8 + \mathrm{O}(t^{10}).$$

Thus, we see that we have one quadratic and three quartic invariants. The latter are generated by $\delta_{ijkl}$, $\zeta_{ijkl}$ and $\xi_{ijkl}$, and their form is given in (22a,b) and (10) with $m = 2$. The unique quadratic invariant is obviously generated by $\delta_{ij}$ and it is given by $\phi^2 = \phi_1^2 + \phi_2^2 + \cdots + \phi_{2n}^2$.

## 4.2 Projectors and crossing equation

If we now define

$$\widehat{P}^S_{ijkl} = \frac{1}{2n}\delta_{ij}\delta_{kl}, \tag{28a}$$

$$\widehat{P}^W_{ijkl} = \frac{1}{2}(\delta_{ijkl} - \zeta_{ijkl}) - \frac{1}{3}\omega_{ijkl}, \tag{28b}$$

---

[11]For $n = 3, 4$ the computation of the Molien series was performed with GAP [32].

$$\widehat{P}^X_{ijkl} = \tfrac{1}{2}(\delta_{ijkl} + \zeta_{ijkl}) + \tfrac{1}{3}\omega_{ijkl} - \tfrac{1}{2n}\delta_{ij}\delta_{kl}, \tag{28c}$$

$$\widehat{P}^Y_{ijkl} = \zeta_{ijkl} - \tfrac{1}{3}\omega_{ijkl}, \tag{28d}$$

$$\widehat{P}^Z_{ijkl} = -\delta_{ijkl} - \zeta_{ijkl} + \tfrac{1}{3}\omega_{ijkl} + \tfrac{1}{2}(\delta_{ik}\delta_{jl} + \delta_{il}\delta_{jk}), \tag{28e}$$

$$\widehat{P}^A_{ijkl} = \tfrac{1}{3}(\omega_{ijkl} + 2\omega_{ikjl}), \tag{28f}$$

$$\widehat{P}^B_{ijkl} = -\tfrac{1}{3}(\omega_{ijkl} + 2\omega_{ikjl}) + \tfrac{1}{2}(\delta_{ik}\delta_{jl} - \delta_{il}\delta_{jk}), \tag{28g}$$

we may verify, using (23) and (24), the projector relations

$$\widehat{P}^I_{ijmn}\widehat{P}^J_{mnkl} = \widehat{P}^I_{ijkl}\,\delta^{IJ}, \qquad \sum_I \widehat{P}^I_{ijkl} = \delta_{ik}\delta_{jl}, \qquad \widehat{P}^I_{ijkl}\delta_{ik}\delta_{jl} = \hat{d}^I_r, \tag{29}$$

where $\hat{d}^I_r$ is the dimension of the representation indexed by $I$, with

$$\{\hat{d}^S_r, \hat{d}^W_r, \hat{d}^X_r, \hat{d}^Y_r, \hat{d}^Z_r, \hat{d}^A_r, \hat{d}^B_r\} = \{1, n, n-1, n, 2n(n-1), n, 2n(n-1)\}. \tag{30}$$

The generalization of (21), valid for any $n \geqslant 2$, is

$$\overset{2n}{\phi_i} \times \overset{2n}{\phi_j} \sim \delta_{ij}\overset{1}{S} + \overset{n}{W}_{(ij)} + \overset{n-1}{X}_{(ij)} + \overset{n}{Y}_{(ij)} + \overset{2n(n-1)}{Z}_{(ij)} + \overset{n}{A}_{[ij]} + \overset{2n(n-1)}{B}_{[ij]}. \tag{31}$$

The projectors (28a–g) allow us to express the four-point function of interest in a conformal block decomposition in the $12 \to 34$ channel:

$$\langle \phi_i(x_1)\phi_j(x_2)\phi_k(x_3)\phi_l(x_4) \rangle = \frac{1}{(x_{12}^2 x_{34}^2)^{\Delta_\phi}} \sum_I \sum_{\mathcal{O}_I} \lambda^2_{\mathcal{O}_I} \widehat{P}^I_{ijkl}\, g_{\Delta_I,\ell_I}(u,v), \tag{32}$$

where the sum over $I$ runs over the representations $S, W, X, Y, Z, A, B$. For the crossing equation we find[12]

$$
\sum_{S^+} \lambda^2_{\mathcal{O}} \begin{pmatrix} F^-_{\Delta,\ell} \\ 0 \\ 0 \\ 0 \\ 0 \\ F^+_{\Delta,\ell} \\ F^+_{\Delta,\ell} \end{pmatrix}
+ \sum_{W^+} \lambda^2_{\mathcal{O}} \begin{pmatrix} 0 \\ F^-_{\Delta,\ell} \\ 0 \\ 0 \\ 0 \\ -F^+_{\Delta,\ell} \\ 0 \end{pmatrix}
+ \sum_{X^+} \lambda^2_{\mathcal{O}} \begin{pmatrix} -\tfrac{1}{n}F^-_{\Delta,\ell} \\ F^-_{\Delta,\ell} \\ F^-_{\Delta,\ell} \\ -F^-_{\Delta,\ell} \\ 0 \\ (1-\tfrac{1}{n})F^+_{\Delta,\ell} \\ -\tfrac{1}{n}F^+_{\Delta,\ell} \end{pmatrix}
+ \sum_{Y^+} \lambda^2_{\mathcal{O}} \begin{pmatrix} 0 \\ 0 \\ F^-_{\Delta,\ell} \\ 0 \\ 0 \\ -F^+_{\Delta,\ell} \\ 0 \end{pmatrix}
$$

$$
+ \sum_{Z^+} \lambda^2_{\mathcal{O}} \begin{pmatrix} 2F^-_{\Delta,\ell} \\ -2F^-_{\Delta,\ell} \\ -2F^-_{\Delta,\ell} \\ 2F^-_{\Delta,\ell} \\ F^-_{\Delta,\ell} \\ 0 \\ -F^+_{\Delta,\ell} \end{pmatrix}
+ \sum_{A^-} \lambda^2_{\mathcal{O}} \begin{pmatrix} 0 \\ 0 \\ 0 \\ F^-_{\Delta,\ell} \\ 0 \\ F^+_{\Delta,\ell} \\ 0 \end{pmatrix}
+ \sum_{B^-} \lambda^2_{\mathcal{O}} \begin{pmatrix} 0 \\ 0 \\ 0 \\ 0 \\ F^-_{\Delta,\ell} \\ 0 \\ F^+_{\Delta,\ell} \end{pmatrix}
= \begin{pmatrix} 0 \\ 0 \\ 0 \\ 0 \\ 0 \\ 0 \\ 0 \end{pmatrix}. \tag{33}
$$

---

[12]In (33) we omit, for brevity, to label the $F_{\Delta,\ell}$'s and $\lambda^2_{\mathcal{O}}$'s with the appropriate index $I$. The appropriate labeling, however, is obvious from the overall sum in each term.

Let us make a comment about (33). We observe that we obtain the same crossing equation if we exchange the second and third line in all vectors and at the same time relabel $W^+ \leftrightarrow Y^+$. This implies, for example, that operator dimension bounds on the leading scalar $W$ operator and the leading scalar $Y$ operator will be identical. Furthermore, if we work out the spectrum on the $W$- and the $Y$-bound, then all operators in the solution will have the same dimensions in both cases (except for the relabeling $W^+ \leftrightarrow Y^+$). The reason for this is that there exists a transformation of $\phi_i$ that permutes the projectors $\widehat{P}^W$ and $\widehat{P}^X$.[13] Indeed, if

$$\phi_i \to \tfrac{1}{\sqrt{2}}(\phi_i + \phi_{i+1}), \quad i \text{ odd} \quad \text{and} \quad \phi_i \to \tfrac{1}{\sqrt{2}}(\phi_{i-1} - \phi_i), \quad i \text{ even}, \tag{34}$$

then

$$\delta_{ij} \to \delta_{ij}, \quad \delta_{ijkl} \to \tfrac{1}{2}(\delta_{ijkl} + 3\zeta_{ijkl}), \quad \zeta_{ijkl} \to \tfrac{1}{2}(\delta_{ijkl} - \zeta_{ijkl}) \quad \text{and} \quad \omega_{ijkl} \to \omega_{ijkl}. \tag{35}$$

Under (35) we obviously have $\widehat{P}^W \leftrightarrow \widehat{P}^Y$. Let us remark here that something similar happens in the $N = 2$ cubic theory studied in [9, Sec. 6], again due to the transformation (34) that exchanges two projectors.[14]

With the crossing equation (33) we can now commence our numerical bootstrap explorations. Before that, however, let us first summarize results of the $\varepsilon$ expansion for theories with tetragonal anisotropy.

## 5 Tetragonal anisotropy

Theories with tetragonal anisotropy were first studied with the $\varepsilon$ expansion a long time ago in [19–23] and later [33], and they were revisited recently in [13, 31]. A standard review is [12, Sec. 11.6]. The Lagrangian one starts with is[15]

$$\mathcal{L} = \tfrac{1}{2} \partial_\mu \phi_i \partial^\mu \phi_i + \tfrac{1}{8}(\lambda \xi_{ijkl} + \tfrac{1}{3} g_1 \delta_{ijkl} + \tfrac{1}{3} g_2 \zeta_{ijkl}) \phi_i \phi_j \phi_k \phi_l. \tag{36}$$

For $g_1 = g_2 = g$ this reduces to (20). The theory (36) in $d = 4 - \varepsilon$ has six inequivalent fixed points. They are[16]

1. Gaussian ($\lambda = g_1 = g_2 = 0$),

2. $2n$ decoupled Ising models ($\lambda = g_2 = 0, g_1 > 0$),

3. $n$ decoupled $O(2)$ models ($\lambda = 0, g_1 = g_2 > 0$),

4. $O(2n)$ ($\lambda > 0, g_1 = g_2 = 0$),

5. Hypercubic with symmetry $C_{2n} = \mathbb{Z}_2^{2n} \rtimes S_{2n}$ ($\lambda > 0, g_1 > 0, g_2 = 0$),[17]

6. $n$ coupled $O(2)$ models with symmetry $MN_{2,n} = O(2)^n \rtimes S_n$ ($\lambda > 0, g_1 = g_2 > 0$).[18]

Note that in the $\varepsilon$ expansion there is no $R_n$ symmetric fixed point. According to the $\varepsilon$ expansion the stable fixed point is the $MN_{2,n}$ symmetric one we discussed in section 3.

---

[13]This was suggested to us by Hugh Osborn.

[14]In the $N = 2$ cubic case, which corresponds to $n = 1$ here in which case the $\zeta$ tensor does not exist, we can show that $\delta_{ij} \to \delta_{ij}$ and $\delta_{ijkl} \to -\delta_{ijkl} + \tfrac{1}{2}(\delta_{ij}\delta_{kl} + \delta_{ik}\delta_{kl} + \delta_{il}\delta_{jk})$.

[15]Compared to couplings $\lambda, g_1, g_2$ of [13, Sec. 7] we have $\lambda^{\text{here}} = \lambda^{\text{there}} - \tfrac{2}{3(n+1)} g^{\text{there}}$, $g_1^{\text{here}} = g_1^{\text{there}}$ and $g_2^{\text{here}} = g_2^{\text{there}}$.

[16]Fixed points physically-equivalent to those in items 2 and 5 on the list are also found in other positions in coupling space, related to the ones given in the list by the field redefinition in (34) [12, 13].

[17]The theory of $2n$ decoupled Ising models in item 2 on the list has symmetry $C_{2n}$ as well. However, we reserve the $C_{2n}$ characterization for the theory in 5.

[18]The theory of $n$ decoupled $O(2)$ models in item 3 on the list has symmetry $MN_{2,n}$ as well. However, we reserve the $MN_{2,n}$ characterization for the theory in item 6.

# 6 Numerical results

The numerical results in this paper have been obtained with the use of PyCFTBoot [26] and SDPB [34]. We use $nmax = 9$, $mmax = 6$, $kmax = 36$ in PyCFTBoot and we include spins up to $\ell_{max} = 26$. For SDPB we use the options -findPrimalFeasible and -findDualFeasible and we choose precision $= 660$, dualErrorThreshold $= 10^{-20}$ and default values for other parameters.

## 6.1 MN

For theories with $MN_{m,n}$ symmetry the bound on the leading scalar singlet is the same as the bound on the leading scalar singlet of the $O(mn)$ model. We will thus focus on bounds on the leading scalar in the $X$ sector, which we have found to display the most interesting behavior. Let us mention here that in the theory of $n$ decoupled $O(m)$ models the dimension of the leading scalar in the $X$ sector is the same as the dimension of the leading scalar in the two-index traceless-symmetric irrep of $O(m)$. Based on the results of [5] we can see that the theory of $n$ decoupled $O(m)$ models is located deep in the allowed region of our corresponding $X$-bounds below.

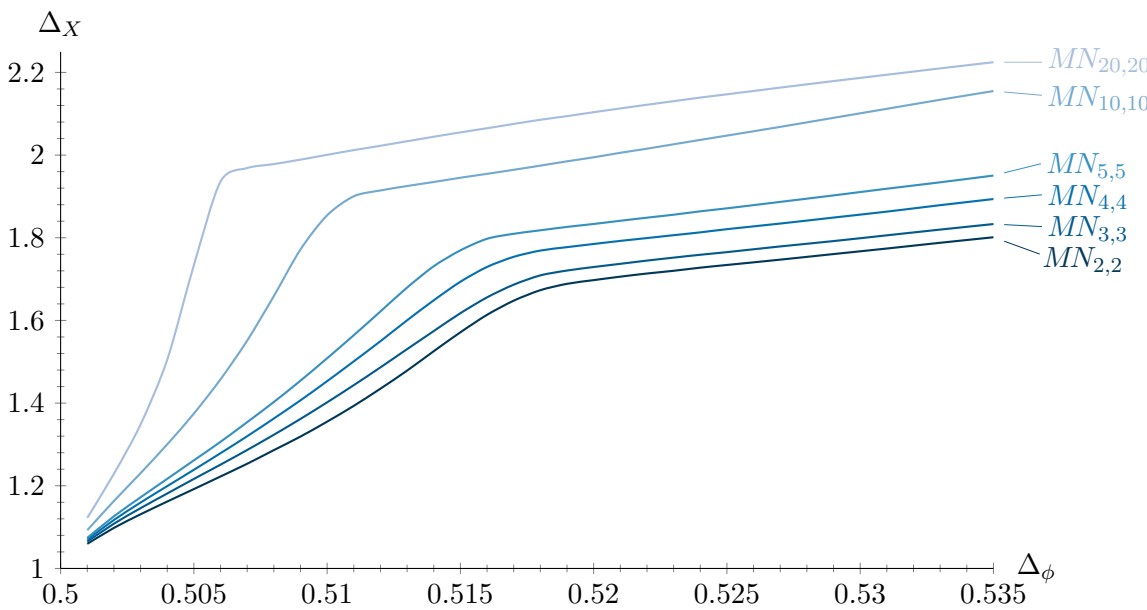

Figure 1: Upper bound on the dimension of the first scalar $X$ operator in the $\phi_i \times \phi_j$ OPE as a function of the dimension of $\phi$. Areas above the curves are excluded in the corresponding theories.

For some theories with $m = n$ the bounds are shown in Fig. 1. The form of these bounds is rather suggestive regarding the large $m, n$ behavior of the $MN_{m,n}$ theories. Recall that in the $O(N)$ models as $N \to \infty$ we have $\Delta_\phi^{O(N)} \to \frac{1}{2}$ and $\Delta_S^{O(N)} \to 2$. There is another case where a type of large $N$ expansion exists, namely in the $O(m) \times O(n)$ theories [35,37]. There, for fixed $m$ one can find a well-behaved expansion at large $n$. Of course $m$ and $n$ are interchangeable in the $O(m) \times O(n)$ example, but in our $MN_{m,n}$ case it is not clear if we should expect the large-$N$ behavior to arise due to $m$ or due to $n$. It is perhaps not surprising that it is in fact due to $m$. Keeping $m$ fixed and increasing $n$ does not have a significant effect on the location of the kink—see Fig. 2. On the other hand, keeping $n$ fixed and raising $m$ causes the kink to move toward the point $(\frac{1}{2}, 2)$—see Fig. 2. (After these bootstrap results were obtained the authors

of [13] realized that the large-$m$ expansion was easy to obtain in the $\varepsilon$ expansion and they updated the arXiv version of [13] to include the relevant formulas. The anomalous dimension of $X$ is equal to $\varepsilon$ at leading order in $1/m$, and so $\Delta_X^{\varepsilon} = d - 2 + \varepsilon + O(\frac{1}{m}) = 2 + O(\frac{1}{m})$.)

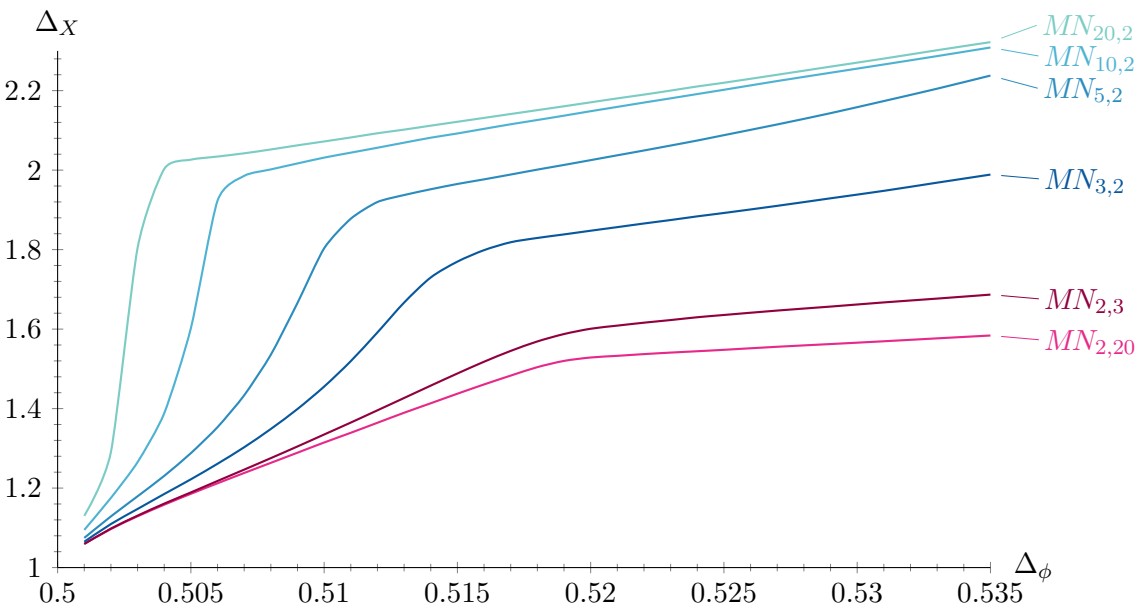

Figure 2: Upper bound on the dimension of the first scalar $X$ operator in the $\phi_i \times \phi_j$ OPE as a function of the dimension of $\phi$. Areas above the curves are excluded in the corresponding theories.

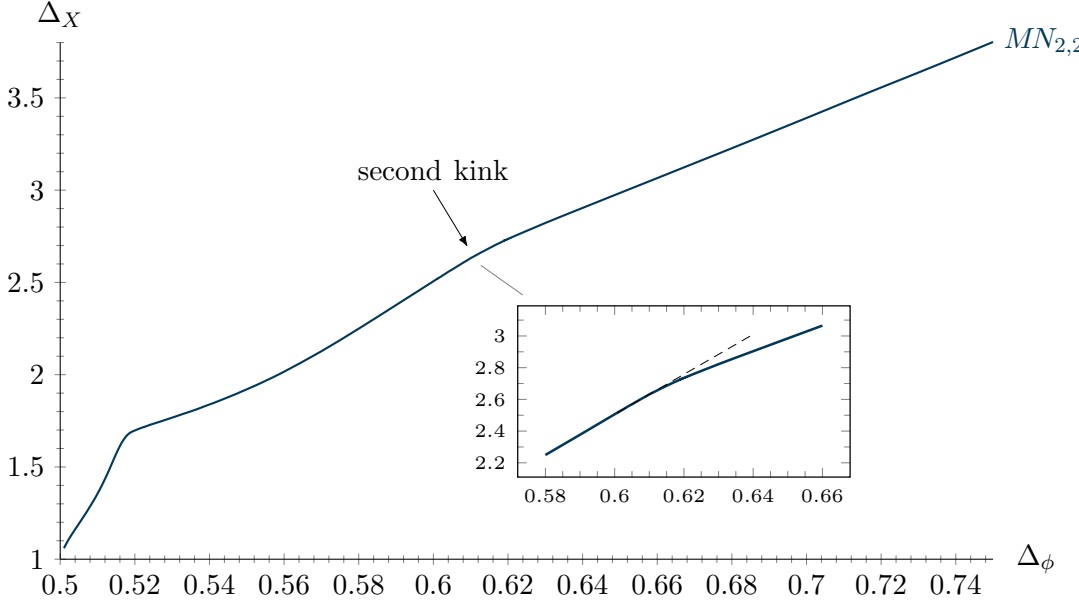

Figure 3: Upper bound on the dimension of the first scalar $X$ operator in the $\phi_i \times \phi_j$ OPE as a function of the dimension of $\phi$ in the $MN_{2,2}$ theory. The area above the curve is excluded.

Continuing our investigation of the $MN_{2,2}$ theory for larger $\Delta_\phi$ we obtain Fig. 3. There we observe the presence of a second kink. Although not as convincing as the kink for smaller $\Delta_\phi$ in the same theory, it is tempting to associate this kink with the presence of an actual CFT.

This is further supported by the results from our spectrum analysis which give us the critical exponents (3) that match experimental results very well as mentioned in the introduction.

Let us mention here that the spectrum analysis consists of obtaining the functional $\vec{\alpha}$ right at the boundary of the allowed region (on the disallowed side) and looking at its action on the vectors $\vec{V}_{\Delta,\ell}$ of $F^{\pm}_{\Delta,\ell}$ that appear in the crossing equation $\sum_{\text{all sectors}} \lambda^2_{\mathcal{O}} \vec{V}_{\Delta,\ell} = -\vec{V}_{0,0}$, where $\vec{V}_{0,0}$ is the vector associated with the identity operator. Zeroes of $\vec{\alpha} \cdot \vec{V}_{\Delta,\ell}$ appear for $(\Delta, \ell)$'s of operators in the spectrum of the CFT that saturates the kink and provide a solution to the crossing equation. More details for this procedure can be found in [16] and [9, Sec. 3.2]. For the determination of critical exponents we simply find the dimension that corresponds to the first zero of $\vec{\alpha} \cdot \vec{V}_{\Delta_S,0}$.

For the $MN_{2,3}$ theory we also find a second kink—see Fig. 4—which is more pronounced than in the $MN_{2,2}$ case. A spectrum analysis for the theory that lives on this second kink yields the critical exponents 4, in good agreement with the measurement of [24].

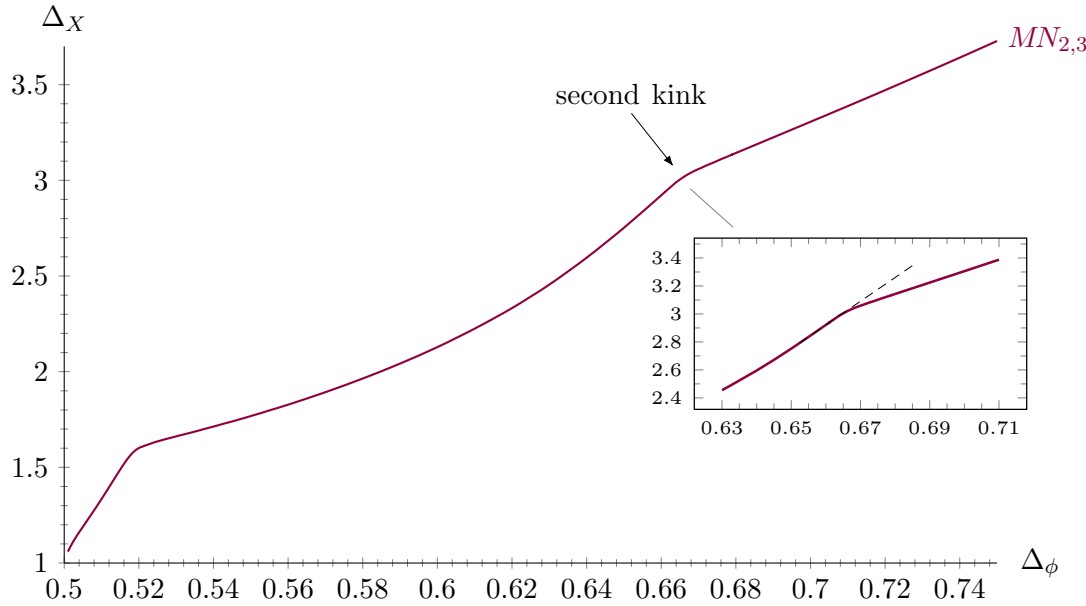

Figure 4: Upper bound on the dimension of the first scalar $X$ operator in the $\phi_i \times \phi_j$ OPE as a function of the dimension of $\phi$ in the $MN_{2,3}$ theory. The area above the curve is excluded.

Another physical quantity one can study in a CFT is the central charge $C_T$, i.e. the coefficient in the two-point function of the stress-energy tensor:

$$\langle T_{\mu\nu}(x) T_{\rho\sigma}(0)\rangle = C_T \frac{1}{S_d^2} \frac{1}{(x^2)^d} \mathcal{I}_{\mu\nu\rho\sigma}(x), \tag{37}$$

where $S_d = 2\pi^{\frac{1}{2}d}/\Gamma(\frac{1}{2}d)$ and

$$\mathcal{I}_{\mu\nu\rho\sigma} = \tfrac{1}{2}(I_{\mu\rho} I_{\nu\sigma} + I_{\mu\sigma} I_{\nu\rho}) - \frac{1}{d}\eta_{\mu\nu}\eta_{\rho\sigma}, \qquad I_{\mu\nu} = \eta_{\mu\nu} - \frac{2}{x^2}x_\mu x_\nu. \tag{38}$$

The central charge of a free scalar in $d = 3$ is $C_T^{\text{scalar}} = \frac{3}{2}$.

In Figs. 5 and 6 we obtain values of the central charge of $MN_{2,2}$ and $MN_{2,3}$ theories assuming that the leading scalar $X$ operator lies on the bound in Figs. 3 and 4, respectively. The free theory of $mn$ scalars has central charge $C_T^{\text{free}} = mn\, C_T^{\text{scalar}} = \frac{3}{2}mn$. We observe two local minima in Figs. 5 and 6, located at $\Delta_\phi$'s very close to those of the kinks in Figs. 3 and 4. We

consider this a further indication of the existence of the CFTs we have associated with the kinks in Figs. 3 and 4.

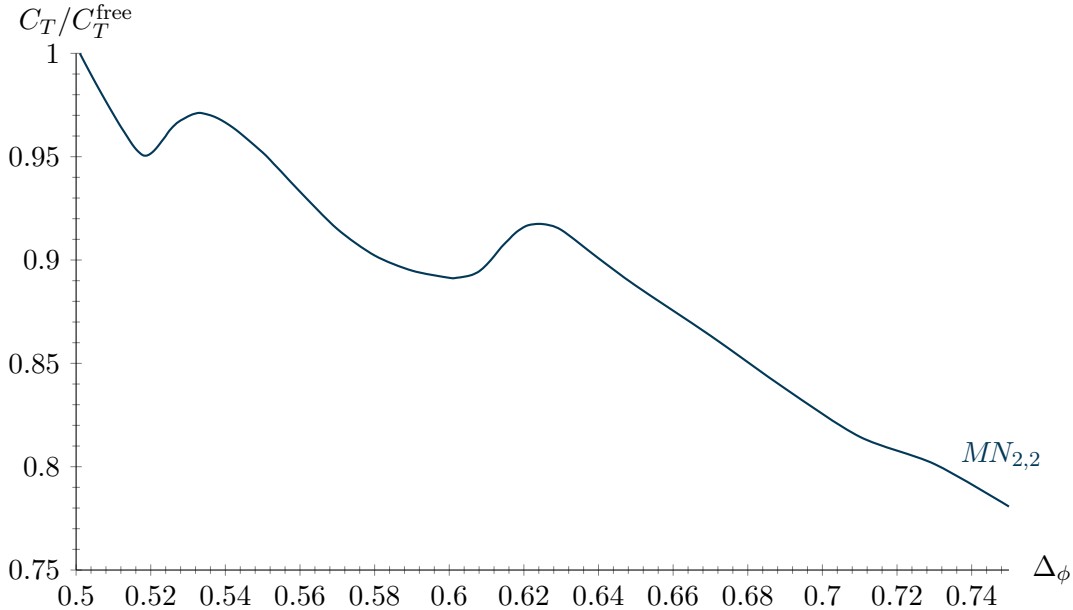

Figure 5: Central charge values in the $MN_{2,2}$ theory assuming that the dimension of the leading scalar $X$ operator lies on the bound in Fig. 3.

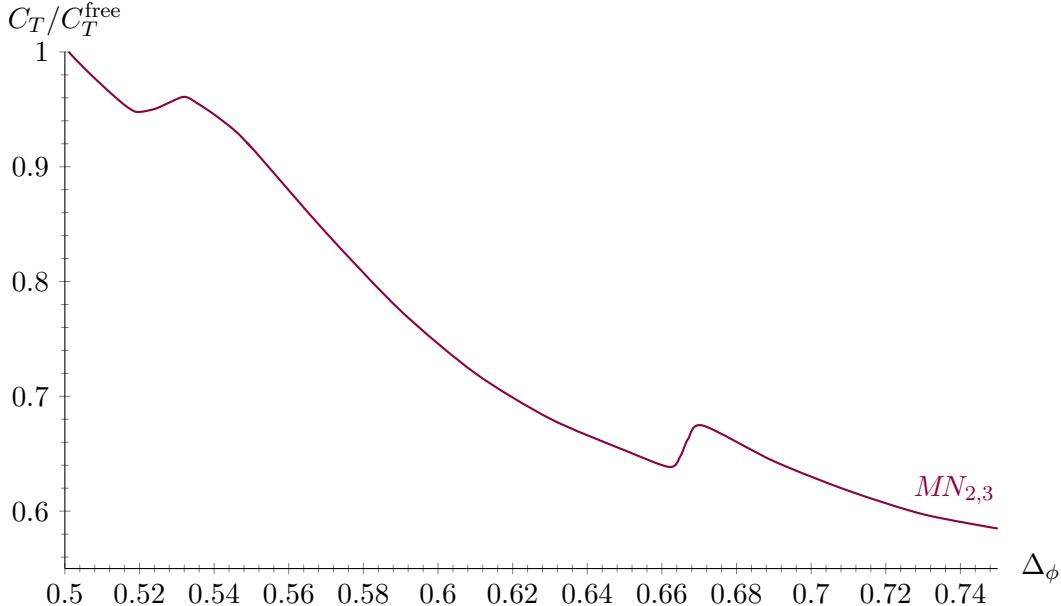

Figure 6: Central charge values in the $MN_{2,3}$ theory assuming that the dimension of the leading scalar $X$ operator lies on the bound in Fig. 4.

## 6.2 Tetragonal

The bound on the leading scalar in the singlet sector in the $R_n$ theory is identical, for the cases checked, to the bound obtained for the leading scalar singlet in the $O(2n)$ model. The bound on the leading scalar in the $X$ sector is identical, again for the cases checked, to the bound on

the leading scalar in the $X$ sector of the $MN_{2,n}$ theory. Both these symmetry enhancements are allowed, and they show that if a tetragonal CFT exists, then its leading scalar singlet operator has dimension in the allowed region of the bound of the leading scalar singlet in the $O(2n)$ model. A similar comment applies to the leading scalar $X$ operator and the bound on the leading scalar $X$ operator of the $MN_{2,n}$ theory.

Let us focus on the bound of the leading scalar in the $W$ sector, shown in Fig. 7. It turns

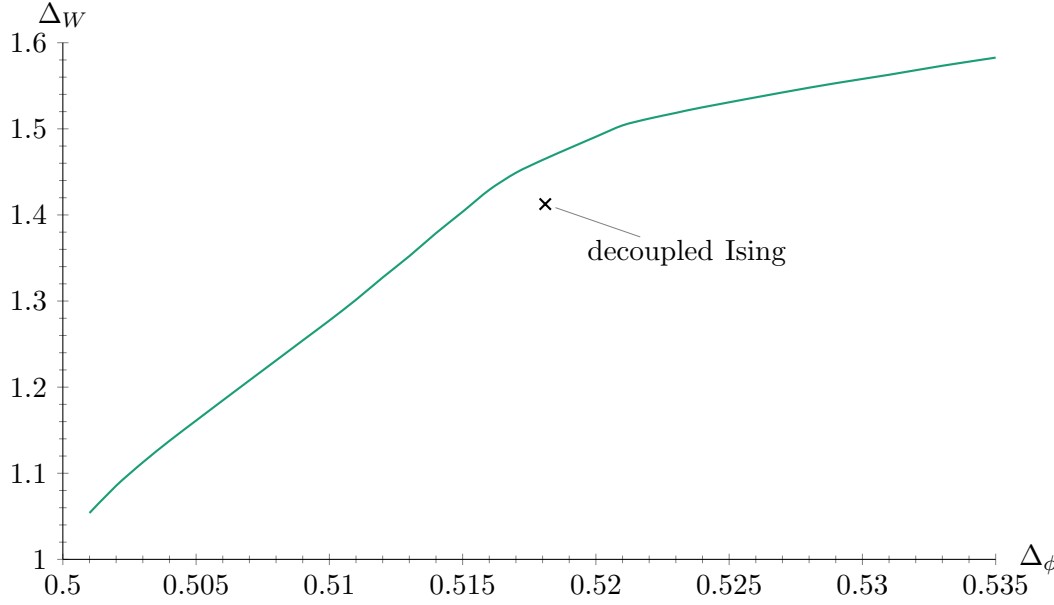

Figure 7: Upper bound on the dimension of the first scalar $W$ operator in the $\phi_i \times \phi_j$ OPE as a function of the dimension of $\phi$. The area above the curve is excluded. This bound applies to all $R_n$ theories checked.

out that the $W$-bound is the same for all $n$ checked, even for $n$ very large. It is also identical to the $V_\square$-bound in [9, Fig. 14]. The coincidence of the $W$ bound with that of [9, Fig. 14] is ultimately due to the fact that the $N = 2$ "cubic" theory has global symmetry $D_4$. Indeed, taking $n$ decoupled copies of the $D_4$ theory leads to a theory with symmetry $R_n$. The leading scalar operator $V_\square$, whose dimension is bounded in the $D_4$ theory in [9, Fig. 14], gives rise to a direct-sum representation that is reducible under the action of $R_n$. That representation splits into two irreps of $R_n$, namely our $W$ and $X$, and it is easy to see that, if the $R_n$ theory is decoupled, the leading scalar operator in the irrep $W$ must have the same dimension as $V_\square$ of [9, Fig. 14]. Hence, the corresponding bounds have a chance to coincide and indeed they do. If a fully interacting $R_n$ theory exists, then the dimension of the leading scalar $W$ operator of that theory is in the allowed region of Fig. 7. We point out here that the putative theory that lives on the bound of [9, Fig. 14] is not predicted by the $\varepsilon$ expansion. That theory is currently under investigation with a mixed-correlator bootstrap [36].

To see if a fully-interacting $R_n$ theory exists, we have obtained bounds for the leading scalar and spin-one operators in other sectors. Unfortunately, our (limited) investigation has not uncovered any features that could signify the presence of hitherto unknown CFTs with $R_n$ global symmetry.

# 7 Conclusion

In this paper we have obtained numerical bootstrap bounds for three-dimensional CFTs with global symmetry $O(m)^n \rtimes S_n$ and $D_4{}^n \rtimes S_n$, where $D_4$ is the dihedral group of eight elements. The $O(m)^n \rtimes S_n$ case displays the most interesting bounds. We have found clear kinks that appear to correspond to the theories predicted by the $\varepsilon$ expansion and have observed that the $\varepsilon$ expansion appears to be unsuccessful in predicting the critical exponents and other observables with satisfactory accuracy in the $\varepsilon \to 1$ limit.

Experiments in systems that are supposed to be described by CFTs with $O(2)^2 \rtimes S_2$ symmetry have yielded two sets of critical exponents [15]. Having found two kinks in a certain bound for such CFTs, we conclude that there are two distinct universality classes with $O(2)^2 \rtimes S_2$ global symmetry. Our critical-exponent computations in these two different theories, given in (1) and (3), match very well the experimental results. It would be of great interest to examine further the conditions under which the renormalization-group flow is driven to one or the other CFT.

For theories with $O(2)^3 \rtimes S_3$ symmetry we also find two kinks. The corresponding critical exponents are given in (2) and (4). The CFT that lives on the second kink, with critical exponents (4), is the one with which we can reproduce experimental results. This is not the CFT predicted by the $\varepsilon$ expansion. A more complete study of the second set of kinks that appear in our bounds would be of interest. Note that the kinks we find do not occur in dimension bounds for singlet scalar operators, so we consider it unlikely (although we cannot exclude it) that the second kinks correspond to a theory with a different global symmetry group as has been observed in a few other cases [38–40].

Beyond the examples mentioned or studied in this work, bootstrap studies of CFTs with $O(2) \times O(N)$ and $O(3) \times O(N)$ symmetry with $N > 2$ have been performed in [41, 42], where evidence for a CFT not seen in the $\varepsilon$ expansion was presented. Such CFTs have been suggested to be absent in perturbation theory but arise after resummations of perturbative beta functions. Examples have been discussed in $O(2) \times O(N)$ frustrated spin systems [43–45]. These examples have been criticized in [46, 49]. However, the results of [42] for the $O(2) \times O(3)$ case are in good agreement with those of [43, 45], lending further support to the suggestion that new fixed points actually exist. Our results (3) are also in good agreement with the corresponding determinations of critical exponents in [43, 45].

The study of more examples with numerical conformal bootstrap techniques is necessary in order to examine the conditions under which perturbative field theory methods may fail to predict the presence of CFTs or in calculating the critical exponents and other observables with accuracy. Examples of critical points examined with the $\varepsilon$ expansion in [12, 13, 31, 47] constitute a large unexplored set. The generation of crossing equations for a wide range of finite global symmetry groups was recently automated [48]. This provides a significant reduction of the amount work required for one to embark on new and exciting numerical bootstrap explorations.

# Acknowledgements

I am grateful to Hugh Osborn for countless important comments, remarks and suggestions, and Slava Rychkov and Alessandro Vichi for many illuminating discussions. I also thank Kostas Siampos for collaboration in the initial stages of this project. Some computations in this paper have been performed with the help of *Mathematica* and the package xAct. The numerical computations in this paper were run on the LXPLUS cluster at CERN.

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
