# Peer review of "Bootstrapping MN and Tetragonal CFTs in Three Dimensions"

_SciPost Physics, doi:SciPost Phys. 7, 010 (2019)_

## Round 1 · Referee Report · Anonymous · 2019-5-29

Strengths

The paper
1-is well written and clear
2-addresses a very interesting problem
3-clarifies some discrepancies, observed in the literature
4-has potential connection with experiments

Weaknesses

1- Parts of the paper are very technical
2-Quite difficult to be accessed by non experts

Report

This paper addresses an interesting problem which is the application of numerical bootstrap techniques to conformal field theories with specific global symmetry gauge groups (semidirect products of $K^n \rtimes S_n$). Such cases are interesting both because beside the $\epsilon$ expansion there are no other methods to study these theories and also because they are central in the description of phase transitions in helimagnets and antiferromagnets. The paper shed light on the existence of such theories in three dimensions and on the values of critical exponents, for some specific theories. The results are quite robust and promising, despite the fact that it is the first time that numerical conformal bootstrap techniques are applied to these systems. This paper deserves the publication, after few points have been clarified (see section ''requested changes'')

Requested changes

1- I think it would be better if the author reports also the experimental data in a table, which could be displayed in the introduction.
2-It is not very clear to me the contradiction with refs [18,12] of the paper's reference list discussed in one of the last paragraphs of the introduction. Would it be possible that the CFT that saturates the bound is just different? meaning has one critical exponent which is the same (what is denoted by $\nu$) but not the other ($\beta$)?
3- It would be better to improve readability to have some of the details of sec 2.1 and 4.1 in an appendix

  • validity: top
  • significance: high
  • originality: high
  • clarity: top
  • formatting: excellent
  • grammar: excellent

Author:  Andreas Stergiou  on 2019-06-05  [id 532]

(in reply to Report 1 on 2019-05-29)

I would like to thank the referee for their careful reading of my manuscript. My reply here includes a list of changes I have made to my manuscript in response to the referee's "Requested changes". I hope they will be deemed appropriate and sufficient by the referee.

The referee's suggestion (1), namely to add experimental results in the introduction, is much appreciated. I have updated the arXiv version of the manuscript (to v2) to include experimental and $\varepsilon$ expansion results in the introduction. I decided to include those results in the text where appropriate, as opposed to listing them in a table. The only results not listed in my paper, and to which the reader is referred to, are now those of table 37 of the review by Pelissetto and Vicari, reference [9] in the manuscript. I did not think it was appropriate to reproduce (copy) that table in my manuscript.

To address the referee's suggestion (2), I added a sentence in the relevant paragraph explaining that I attribute the presence of kinks to second-order phase transitions. The suggestion of references [18, 12] is that there are no second-order phase transitions, but rather that the experimental results are explained by weakly first-order phase transitions.

Regarding the referee's suggestion (3), I think that the equations presented in the main text in sections 2.1 and 4.1 are important as they elucidate the way the crossing equations have been obtained. Although one could argue that the method used is standard practice, I still believe that it is worth emphasizing the relevant tensors and equations, for they provide essential guidance when trying to apply the bootstrap program to other cases of interest.

Anonymous on 2019-06-13  [id 540]

(in reply to Andreas Stergiou on 2019-06-05 [id 532])
Category:
remark
correction

I would like to bring to your attention a new arXiv version of my manuscript (v3), where I corrected some erroneous statements regarding the applicability of my bootstrap results to stacked triangular antiferrromagnets. In particular, I removed such statements from the introduction, and added a paragraph in section 3 describing why my $MN_{2,2}$ results do not actually apply to XY stacked triangular antiferromagnets. Numerous other applications remain, so this does not diminish the relevance of my results for systems of interest.

---

## Editorial Decision

published